# Effect of Treatment Methods on Material Properties and Performance of Sawdust-Concrete and Sawdust-Polymer Composites

**DOI:** 10.3390/polym16233289

**Published:** 2024-11-26

**Authors:** Arafater Rahman, Mohammad Abu Hasan Khondoker

**Affiliations:** Industrial System Engineering, University of Regina, 3737 Wascana Pkwy, Regina, SK S4S, Canada; arf225@uregina.ca

**Keywords:** sawdust concrete, sawdust polymer composite, chemical treatment, tensile strength, compressive strength, microstructural examination

## Abstract

The circular economic approach in polymer composite research has gained acceptance for offering low-cost, high-performance solutions. Sawdust-derived composites have drawn interest as alternatives in concrete and composite fabrication, addressing housing shortages and resource depletion. Sawdust concrete (SDC) and sawdust polymer composites (SDPC) are key areas under investigation, with SDC additionally aiding in carbon reduction in building materials. However, challenges arise due to sawdust’s inherent hydrophilicity, porosity, and lower strength. This study introduces a novel approach by identifying specific chemical treatments, including alkali and silane, which effectively enhance sawdust’s compressive and tensile strengths, moisture resistance, and durability, optimizing it for structural applications. The study evaluates SDC’s compressive strength based on treatment type, concentration, and curing time, examining physical properties such as water absorption, moisture sensitivity, and fiber-matrix adhesion. The unique contribution lies in a detailed optimization analysis, revealing conditions under which sawdust reaches structural-grade performance, expanding its potential in sustainable construction. For SPDC, tensile strength improvements are assessed under various chemical compositions, showing that specific polymers form stronger fiber-matrix bonds for greater stability. Morphological studies further explore fiber-matrix compatibility, hydrophobicity, and failure mechanisms. By advancing the understanding of treatment efficacy, this review positions sawdust as a viable, low-cost material alternative, establishing a foundation for sustainable innovation in construction and bio-composite research. These findings contribute to sawdust’s potential as a practical, eco-friendly building material.

## 1. Introduction

The United Nations estimates that the global population will increase to 9.8 billion by 2050 [1], up from 7.9 billion in 2021 [2]. This rapid growth presents significant challenges for communities worldwide, not just in terms of population size but also in the strain it places on resources and infrastructure, particularly in densely populated areas [3,4]. The rising demand for housing, transportation, energy, and basic services is closely linked to urban migration, as more people move from rural areas to cities in search of better opportunities [5,6,7]. This leads to overcrowding, inadequate housing, and overburdened public services, as existing infrastructure struggles to accommodate the growing population [8]. The housing crisis, in particular, becomes a critical issue, as urban areas face increased pressure to provide affordable and adequate homes. However, traditional construction methods and housing solutions have been unable to keep up with this growing demand [9]. Addressing the needs of a rapidly growing population using traditional construction methods, particularly those that depend on materials like concrete, presents several challenges. The rising global demand for construction materials has created significant environmental and economic challenges, especially in the production of conventional materials like concrete and steel [10]. Manufacturing these materials is highly energy-intensive, contributing substantially to greenhouse gas emissions and environmental harm [11]. Concrete production heavily relies on raw materials such as limestone, clay, and water, as well as construction aggregates like sand, gravel, crushed stone, slag, and recycled concrete. To meet the increasing demand for housing, the extraction of these natural resources has intensified, leading to their rapid depletion [12]. Globally, nearly 11 billion tons of natural materials are extracted annually from riverbeds, lakes, and other sources [13]. This resource extraction process often results in habitat destruction, soil degradation, and significant greenhouse gas emissions. In response to these environmental impacts, many countries have imposed restrictions on sand extraction from natural reservoirs like rivers and canals [14,15]. Additionally, the energy-intensive processes involved in concrete production, including transportation and curing, further contribute to environmental harm and climate change [16]. The transportation of large volumes of concrete also significantly increases CO_2_ emissions [17]. The conventional housing market faces significant challenges in meeting the needs of lower-income populations [18,19,20]. Rising land prices, increasing construction costs, and a shortage of available housing units have left many individuals and families struggling to find affordable housing [18,21]. Innovative and long-term solutions are needed to solve the housing shortage as the world’s population rises. Researchers, policymakers, and urban planners are increasingly recognizing the need for eco-friendly and economically viable alternatives to traditional housing. Sustainable housing approaches could involve the use of nontraditional construction materials that are less environmentally harmful. Traditional building methods are highly resource-intensive [22], but these alternative materials offer a more affordable and sustainable solution [23]. Material researchers are exploring bio-inspired materials as potential substitutes for conventional concrete, with sawdust emerging as a promising option due to its widespread availability [24,25]. Sawdust is a rich resource that may be utilized as a bioenergy feedstock, decreasing the environmental impact of fossil fuel consumption and trash disposal. According to research, sawdust may be efficiently used in biofuel production using a variety of conversion processes, including biodegradation and biofuel fermentation [26]. Proper disposal techniques can result in a better environment and economic benefits [27], which are consistent with sustainable waste management. Sawdust may also be utilized in biofuel production, helping to generate biomass energy and increasing soil carbon stores for remediation [28]. It may also be utilized in solid biofuels like pellets and briquettes, which are a greener alternative to conventional fossil fuels. Sawdust’s thermal qualities make it an interesting choice for energy generation since it can cut greenhouse gas emissions. Meanwhile, the disposal of wood waste, particularly sawdust, poses its own environmental concerns, contributing to soil and water pollution [29]. Integrating sawdust into polymer composites and concrete offers a promising approach to addressing these issues. As an abundant, low-cost, and renewable resource, sawdust is an appealing alternative that can enhance the properties of these materials, supporting both environmental sustainability and resource efficiency [30]. For example, over 48 million tonnes of wood waste are generated annually in the EU28 from various sources such as construction, agriculture, and railway sectors, according to EUROSTAT [31]. A significant portion of this waste is found to be contaminated with harmful chemicals previously used to protect wood from decay and insects. Recent studies [32,33] have highlighted the environmental and health risks associated with wood preservatives. In this context, using sawdust as a resource offers a solution to mitigate pollution caused by these chemicals [34]. To create usable products from wood waste, modifying sawdust is a necessary first step. This can be achieved through a range of methods, including drying, grinding, milling, compaction, chemical and heat treatments, adjusting fiber orientation, modifying lignin, using adhesives and binders, controlling particle size, reinforcing with materials like fiberglass, and producing nanocellulose [35,36,37,38]. The method chosen depends on factors like the intended application, cost, and environmental impact [39]. The main engineering challenges of incorporating sawdust into composites involve achieving sufficient interfacial bonding between the sawdust and the polymer. Inadequate adhesion can cause sawdust particles to agglomerate, which weakens the mechanical properties [40,41]. The utilization of chemical treatment is highly appealing in the enhancement of sawdust qualities due to its versatility and capacity to customize the material to fulfill specific requirements [42]. Chemical treatments are particularly attractive for enhancing sawdust properties due to their versatility and ability to tailor the material for specific needs [43]. Unlike physical treatments, which may be limited by sawdust’s natural properties, chemical treatments allow for significant improvements in strength, durability, and resistance to external factors [44,45]. However, to ensure sustainability, it is essential to use eco-friendly chemicals and carefully consider the environmental impacts of these treatments.

This review looks at the impacts of various chemical treatments on sawdust, focusing on aspects such as concentration, curing time, durability, and ambient conditions, in order to improve its use in concrete structures and polymer composites. The evaluation for sawdust concrete (SDC) analyzes treatments such as NaOH and Ca(OH)_2_, highlighting their influence on compressive strength, durability, and water resistance. The focus is on ideal curing conditions. Treatments such as NaOH and benzoyl peroxide are investigated for their effects on tensile strength and fiber–matrix bonding in sawdust polymer composites (SDPC), with FTIR and SEM results supporting the findings. This synthesis gives important insights into improving treatment procedures for long-term uses in building and industry.

## 2. Chemical Treatments

Researchers showed gigantic interest in different chemical treatment techniques, and their impact on sawdust is summarized in Appendix A.

### 2.1. NaOH Treatment

Priya et al. [46] used a NaOH solution to treat sawdust, first ensuring proper preparation of the sawdust powder by using a 1.18 mm sieve, as sawdust from various hard and softwoods was collected in different sizes. To remove impurities and achieve a homogeneous mixture, the sawdust was further sieved through 10 and 20 mesh screens before being mixed. The sawdust was then treated with NaOH solutions at concentrations of 1%, 3%, and 5% wt./v, stirred at room temperature. After treatment, the alkali-treated sawdust was thoroughly washed with water and dried in an oven at 70 °C. It was later used as a 5%, 10%, and 15% replacement for sand in concrete mixtures. Additionally, a chemical analysis of the sawdust showed it contained 34.58% SiO_2_ (Table 1), a primary component of sand, highlighting the potential of sawdust as a suitable sand substitute.

Hossain et al. [47] investigated the effects of alkali treatment on different types of sawdust derived from locally available species in Bangladesh, including Garjon (*Dipterocarpus turbinatus*), Gamari (*Gmelina arborea*), and Kerosene (*Cordia subcordata*). They used coarse and medium sawdust particles of 250 µm and 500 µm, respectively. Each type of sawdust was individually mixed with a 10% NaOH solution for 1.5 h at room temperature. Afterward, the sawdust was neutralized using distilled water until the pH reached 7.0, followed by drying in an oven at 110 °C without interruption. In contrast, Haque et al. [48] conducted a study using a 4% alkali treatment on Swietenia mahagoni sawdust, one of the most common species in Bangladesh, and compared the mechanical properties of untreated and alkali-treated sawdust composites with unsaturated polyester (UP) resin. Priya et al. [49] conducted an experiment to determine the optimal NaOH concentration for treating sawdust used in sawdust-high-density-polypropylene composites. Unlike previous studies that focused on specific types of wood sawdust, they collected randomly mixed sawdust from local sawmills with various shapes. To ensure uniformity, the sawdust was sieved into 1.25 µm particles. Before treatment, the sawdust was washed with distilled water and dried at 65 °C to remove surface dust, ensuring accurate results. The dried sawdust was then submerged in 1%, 3%, 5%, and 7% NaOH solutions for 2 h at room temperature. After treatment, the sawdust was washed with acetic acid and distilled water to neutralize the pH, followed by drying in an oven at 70 °C for three days. Similarly, [50] examined the effect of alkali treatment time on building materials using sawdust from a local furniture shop. After sieving the sawdust to a uniform 1 mm size, they conducted three sets of experiments. In the first set, sawdust was treated with 1% to 6% NaOH solutions at 1% intervals in boiling water, with treatment times ranging from 100 to 160 min in 20 min intervals. This set focused on wood–concrete without recycled paper. In the second set, sawdust was treated with 2% to 8% NaOH solutions at 2% intervals for 80 min. In the third set, sawdust was immersed in 2% and 4% NaOH solutions, with treatment times varying from 40 to 140 min. After treatment, all sawdust samples were washed with distilled water and dried in an oven at 100 °C for 24 h. In another study, [51] explored alkali-treated sawdust Acrylonitrile Butadiene Styrene (ABS) hybrid composites, using a 5% NaOH solution for sawdust contents ranging from 5% to 20%, added in 5% increments. Raw sawdust was sieved, washed, and dried at 60 °C for 12 h before being mixed with the 5% NaOH solution for 1 h at room temperature. The treated sawdust was then washed thoroughly and dried for 24 h at 60 °C. Similarly, [52] investigated the use of 10% NaOH instead of 5% for sawdust content ranging from 5% to 30% combined with oil bean pod shell (OBPD) at 5% intervals. In this case, the sieving operation was performed only after the NaOH treatment. Another study [53] utilized a 5% NaOH solution to assess the feasibility of developing fire-resistant wood panels using sawdust-unsaturated polyester(UP) resin composites. Before treatment, the sawdust was ground and sieved with a 300 µm mesh. After soaking in 5% NaOH for 12 h to improve compatibility between the sawdust and matrix, the sawdust was washed with distilled water and dried at 80 °C for 3 h. Finally, [54] studied the effect of filler material on the tensile performance of polymer epoxy composites. They used 5% NaOH-treated mahagoni sawdust combined with OBPD in a 1:1 ratio, which was dispersed into epoxy resin up to 30% by volume. The sawdust was mixed with 5% NaOH for 1 h at 27 °C, followed by drying in an oven at 100 °C for 8 h. After drying, the sawdust was sieved with a 300 µm mesh.

Rahman et al. [55] employed a 5% NaOH treatment for their sawdust polypropylene (PP) composites, incorporating 0% to 35% treated sawdust at 5% intervals. However, instead of distilled water, they used a solution of benzene, deionized salt, and soap for washing. Notably, they introduced an ice bath at 5 °C for 10 min during the NaOH soaking process, contrasting with other studies that used boiling water. A key parameter in their study was the use of deionized benzene salt for synthesis before NaOH treatment, which helped accelerate the neutralization process; without this step, the treated sawdust would face delays before the washing stage. Similarly, [47] studied sawdust polypropylene (PP) composites to assess the effect of fiber loading ranging from 10% to 50% in 10% intervals. In this case, the raw sawdust was first dried in open sunlight for two days to remove moisture and sieved with a 1–2 mm mesh. The sieved sawdust was then treated with 5% NaOH for 2 h at room temperature, followed by a neutralization process using distilled water, acetic acid, and distilled water. Afterward, the treated sawdust was dried again in open sunlight for three consecutive days.

### 2.2. NaOH and Others

#### 2.2.1. (NaOH + KOH + H_2_O_2_) Treatment

Kong et al. [56] studied the effect of chemical treatments on sawdust–redmud geopolymer composites, using sawdust concentrations of 10 wt.% to 60 wt.% in 10% intervals. The raw sawdust was first ground with an ultra-centrifugal mill, washed, and dried at 40 °C for 24 h. The sawdust was then treated with a 5% NaOH solution for 24 h at room temperature, washed with tap water to achieve a neutral pH, and dried in an oven at 60 °C for another 24 h. Afterward, the sawdust was immersed in a bleaching solution containing 20% $H_{2}O_{2}$ and 3.5 g/L of KOH for 24 h at room temperature. Finally, it was rinsed with water to reach a pH of 7 and dried again at 60 °C for 24 h.

#### 2.2.2. (NaOH + C_14_H_10_O_4_ (Benzoyl Peroxide)) Treatment

Akter et al. [57] investigated the chemical treatment of teak wood (*Tectona grandis*), locally known as Shegun in Bangladesh, to improve hybrid composites with high-density polyethylene (HDPE) by adding 0% to 45% treated sawdust in 5% intervals. They introduced a novel process involving benzoyl peroxide after the NaOH treatment. The sawdust, randomly collected, was sieved using various mesh sizes (70+, 100+, 120+, 140+) and then washed and dried to create a homogenous mixture. The sawdust was immersed in a 5% NaOH solution for 1 h, followed by repeated washing with tap and distilled water until the pH reached 7, and then dried in an oven at 105 °C for 6 h. For the benzoyl peroxide treatment, they applied a trial-and-error approach, testing different conditions such as reaction time, pH level, concentration of benzoyl peroxide, and temperature variations. After each reaction, they used FTIR analysis to determine the best parameters, eventually concluding that a 6% benzoyl peroxide solution yielded optimal results, which they standardized for the remainder of the experiment.

#### 2.2.3. (NaOH + Dimethyldichlorosilane (DMDCS)+ Polydimethylsiloxane (PDMS)) Treatment

Yang et al. [58] reported the positive impact of hydrophobic agents, such as DMDCS and PDMS, on sawdust chemical treatment. They tested treatment times of 5, 20, and 120 min for concentrations of 2%, 5%, and 8%, respectively, for both agents. The hydrophobic reagents were dissolved in an n-hexane solution, and the treatment was performed in a ventilated cabinet to allow easy emission of n-hexane vapor via a vacuum pump. For neutralization, they used a 0.125% alkali solution and then exposed the treated sawdust to 50 °C heated vacuum air for 12 h.

### 2.3. Ca(OH)_2_ Treatment

A feasibility analysis of the circular economic approach in building construction was conducted to assess the potential use of locally available materials in Sudan, such as sawdust as a replacement for sand in concrete mixtures and sawdust ash as a substitute for cement. For the sawdust treatment, they used water and Ca(OH)_2_ separately for 10% and 20% sawdust additions in the concrete mixture. A comparative analysis was performed to evaluate the mechanical performance of untreated, water-treated, and Ca(OH)_2_ treated sawdust. Additionally, untreated sawdust ash, produced through burning and sieving processes, was considered a cement replacement in concrete mixtures [59]. In another study on the use of sawdust in structural concrete, Saeed [60] adopted a unique chemical treatment approach for sawdust. He used a 20% Ca(OH)_2_ solution as a washing agent to remove substances from the sawdust that could hinder the hardening process of concrete. This initial phase, termed Phase I, involved sieving the sawdust with a 6.3 mm mesh and then boiling it in the Ca(OH)_2_ solution for 1 h. He also recommended the addition of ferrous sulfate to the boiling water, followed by drying the treated sawdust in the open air. Additionally, two waterproofing agents—cutback asphalt (MC-70) and classic varnish (not shown in the figure)—were used as primary chemicals for sawdust treatment, each in a 25% ratio to the sawdust weight, as depicted in Figure 1. Nevertheless, Hewayde et al. [61] implemented a three-step chemical treatment process for sawdust, aiming to use it as a potential sand replacement in concrete mixtures. Sawdust from three different wood types—saj, pine, and cedar—was collected in both coarse and fine forms and mixed to create a homogenous blend. The chemical treatments included distilled water treatment, Ca(OH)_2_ treatment, and a combined Ca(OH)_2_ + CaCl_2_ solution treatment. In the first method, sawdust was immersed in distilled water heated to 50 °C for 1 h, then drained using a sieve for 15 min. For the second method, raw sawdust was soaked in a 5% Ca(OH)_2_ solution at room temperature for 1 h, followed by the same distilled water treatment as in the first method. In the third method, after repeating the second process, CaCl_2_ was added to act as an accelerator to counter the concrete’s setting delay.

### 2.4. Na_2_SiO_3_ Treatment

Siddique et al. [62] examined the effects of incorporating sawdust as a replacement for 5% to 20% of natural sand in concrete mixers, analyzing both Na_2_SiO_3_-treated and water-treated sawdust in terms of their mechanical performance compared to control concrete properties. For the treatment process, sawdust was first sieved through a 4.75 mm mesh and then treated separately with water and sodium silicate for 24 h. The study also assessed the physical properties and particle size distribution of the raw sawdust (Figure 2).

### 2.5. (Ca(OH)_2_ + Na_2_SiO_3_) Treatment

An experimental analysis was conducted on four different types of wood waste (see Figure 3) to evaluate the compatibility of treated sawdust with cement in concrete mixers. All wood sawdust underwent a sieving process using a blade mill to achieve a 4 mm size. Following this, the residual extraction process was performed following the EN 12457 standard, where the sieved sawdust was submerged in deionized water for 24 h while being shaken at 10 rpm. Although the deionized water treatment effectively removed liquid chemicals, an additional filtration step using 0.45 µm filter paper was employed to eliminate solid residues. Mass spectroscopy and gas chromatography tests were then conducted to assess feasibility; however, one type of wood did not meet the necessary properties, which is crucial before its use in composite structures. Consequently, chemical treatments were applied to three types of sawdust using Ca(OH)_2_, 1% sodium silicate, and 10% silicate [63].

### 2.6. NaOCl Treatment

Hassen and Hameed [64] explored three distinct chemical processes for treating sawdust intended for use in concrete compositions, with treated sawdust added in amounts ranging from 10% to 50% in 10% increments. The process began with sieving the sawdust through a 2.36 mm mesh to achieve the appropriate size for subsequent treatments. The treatments included (1) soaking the sawdust in a 15% NaOCl solution without rinsing and (2) treating the sawdust with a 15% NaOCl solution, followed by immersion in water for 24 h and then drying. The 15% solution consisted of 0.75 L of NaOCl mixed with 1 kg of sawdust. The mechanical performance of the treated sawdust concrete (SDC) composites was then compared to that of untreated sawdust composites, as the amount of sawdust added remained consistent across all treatments.

### 2.7. Maleic Anhydride (C_4_H_2_O_3_) Treatment

In their study on sawdust–carbon nanotube hybrid composites, Fozing et al. [65] utilized 4 g of 1-butyl-3-methylimidazolium chloride combined with 0.17 g of sawdust, heating the mixture to 120 °C. The mixture was stirred for 30 min under an argon atmosphere. After this period, 0.7 g of maleic anhydride was added, and stirring continued for an additional 4 h. The resulting solution underwent a washing process with distilled water. The washed sawdust was then centrifuged into five 50 mL aliquots, followed by drying at room temperature for 48 h and in an oven at 105 °C for 2 h.

### 2.8. (HCl + NH_4_OH + Polyethylenimine (PEI)) Treatment

Shulga et al. [66] chose aspen (*Populus tremuloides*) sawdust to investigate the effects of adding hydrolyzed and ammonoxidized sawdust combined with a Polyelectrolyte Complex (LPEC) on mechanical performance. The LPEC, which contains lignin, is believed to enhance adhesion, potentially improving water affinity and increasing the contact angle. To produce hydrolyzed sawdust, the researchers subjected it to hydrolysis with HCl at 60 °C for 5 h. This hydrolyzed sawdust was then mixed with an NH_4_OH solution, to which persulfate ammonia was added. The ammonoxidized sawdust was washed with distilled water, dried, and milled in a ball mill for 15 min at 300 rpm. The milled sawdust was then sieved through a 100 µm mesh.

A total of 7.5 g of the modified, hydrolyzed, and ammonoxidized aspen sawdust, with particle sizes smaller than 100 µm, was mixed with 1 L of polyethyleneimine (PEI) solution and stirred for 24 h. After this stirring period, the hydrolyzed and ammonoxidized particles were washed to achieve a pH of 7. Finally, the washed sawdust was dried in an oven at 60 °C for 48 h, followed by 2 h at 105 °C.

### 2.9. (Ethanol+ NaCl+ H_2_SO_4_) Treatment

In sawdust-cement composites, we examined three types of wood—*T. scleroxylon*, *E. cylindricum*, and *K. gabonensis*—focusing on their density. The primary objective was to identify the most suitable sawdust by comparing the chemical constituents and mechanical properties of the composites. To assess the chemical characteristics, including lignin, holocellulose, ash content, and extractive percentage, each type of sawdust underwent a sequential chemical treatment. Initially, the sawdust was dried for 72 h, milled, and sieved through an 850 µm mesh, which was essential before treatment. For quantifying extractives, the dried sawdust powder was mixed with an ethanol-acetone solution in a 1:2 volume ratio for 4 h, followed by immersion in a 95% ethanol solution for another 4 h. After evaporating the extractives, the remaining solution was placed in an oven at 105 °C for 24 h and allowed to cool to room temperature. The total extractive content was calculated using ASTM D-1105-56 standards, revealing that *K. gabonensis* had the highest extractive content at 9.31%, while *E. cylindricum* and *T. scleroxylon* showed 7.675% and 6.116%, respectively.
(1)$$Total\;Extractive\;Concent\left(\%\right)=\frac{Weight\;of\;beaker\;plus\;extractive-weight\;of\;beaker}{Weight\;of\;air-dried\;wood\;meal}\times 100\%$$

To determine lignin content, the extractive-free sawdust was mixed with 15 mL of cold sulfuric acid (72%) for 2 h and then boiled with distilled water for 4 h. The sample was dried in an oven at 103 °C for 2 h before being cooled. Lignin content was measured using the previously mentioned standard, with *K. gabonensis* exhibiting the highest lignin content at 31.59%, while *T. scleroxylon* had the lowest at 29.9%. For holocellulose content evaluation, 2 g of extractive-free sawdust was combined with 180 mL of distilled water, 8.6 g of NaCl, and 6 mL of acetic acid, then heated at 70 °C for 3 h. The heated sawdust was cooled, filtered, and dried in an oven at 103 °C for 2 h. The results indicated that *E. cylindricum* provided the highest holocellulose content at approximately 59%, compared to 56.38% from *T. scleroxylon* and 57.5% from *K. gabonensis*. A straightforward method for determining ash content involved placing the sawdust in a crucible and heating it in a furnace at 600 °C for 3 h, followed by a cooling period before weighing [67].
(2)$$Lignin\;content\left(\%\right)=\frac{Weight\;of\;oven-dried\;residue}{Weight\;of\;oven-dried\;extractive-Free\;sawdust}\times 100\%$$

### 2.10. (Ethanol +Tolune + H_2_O_2_ +Acetic Acid) Treatment

A comparative study was conducted on delignified versus untreated sawdust composites, focusing on their mechanical, thermal, and morphological properties. Sawdust, whether delignified or untreated, was mixed with unsaturated polyester resin at compositions of 5 wt.%, 10 wt.%, 15 wt.%, and 20 wt.%. Initially, the raw sawdust was sieved through an 850 µm mesh, followed by dewaxing using the Soxhlet method with a 2:1 toluene–ethanol mixture, stirred at 150 °C for 2 h, and subsequently dried. For the delignification process, a bleaching solution was prepared consisting of 23.6 wt.% acetic acid, 6.4 wt.% hydrogen peroxide, 69.5 wt.% water, and 0.5 wt.% titanium (IV) oxide. The mixed solution underwent a heat treatment at 130 °C for 3 h to effectively remove lignin. After treatment, the mixture was washed with distilled water to achieve a neutral pH of 7, and then a drying operation was performed [68].

### 2.11. Potassium Methyl Siliconate (PMS) Treatment

Piao et al. [69] studied wood-fiber–sawdust–plastic-hybrid composites. Researchers examined the effects of chemical treatment on interfacial adhesion between the filler and matrix, as well as dimensional stability and moisture permeability. They selected loblolly pine (*Pinus taeda*) and lodgepole pine (*Pinus contorta*) for both the fiber and sawdust components. The sawdust was treated in a solution where the pH was maintained between 8.5 and 10.5 for 4 to 6 h at room temperature, using 60 g of sawdust mixed with 2200 mL of 0.5 M PMS. After treatment, the sawdust was rinsed five times with deionized water and then dried in an oven at 100 °C for 24 h.

### 2.12. Detersive Solvent Treatment

Experiments on locally available teak sawdust–jute composites, focusing on how chemical treatment and exposure to a cryogenic environment impacted the natural fiber composites, were carried out in [70]. Raw sawdust was first sun-dried for two consecutive days and then sieved to obtain three different densities of sawdust. For the chemical treatment, sawdust was submerged in a 1–2% detersive solvent at 60–70 °C for one hour. After the treatment, the sawdust was rinsed with distilled water and dried in an oven for 1.5 h.

### 2.13. Vinyltriethoxysilane (VTES) and Diethyl Ether Treatment

Kusuktham [71] examined the effects of VTES on hybrid composites made from high-density polypropylene (HDPE), rubberwood sawdust, and calcium carbonate (CaCO_3_). The authors conducted stepwise chemical treatments to analyze the sawdust’s chemical composition. First, 25 g of sawdust and CaCO_3_ were treated with a 2% solution of VTES and diethyl ether, and the mixture was heated at 110 °C for 2 h. The treated sawdust then underwent a series of chemical processes to determine its hemicellulose, cellulose, and lignin content.

For lignin analysis, the TAPPI T235 cm-00 standard was used. The treated sawdust was immersed in 250 mL of 10% NaOH, stirred briefly for 10 s, and then placed in a 25 °C water bath for 60 min. Afterward, a solution of 10 mL of 0.5 N potassium dichromate and 30 mL of sulfuric acid (H_2_SO_4_) was added, and the mixture was left for 15 min before diluting with 50 mL of water. Once cooled, ferrous ammonium sulfate (FeH_8_N_2_O_8_S_2_) was added, and the lignin content was calculated using a standard equation.

For hemicellulose, the authors applied the following Equation (3):(3)$$Hemicellulose\;Content\left(\%\right)=\frac{\left[\left(V_{2}-V_{1}\right)\times N\times 6.85\times 10\right]}{A\times W}$$
where V_1_ = Sawdust titration filtrate (mL); V_2_ = blank titration (mL); N= normality of FeH_8_N_2_O_8_S_2_; A = filtrated sawdust volume in oxidation (mL); and W = sawdust dry weight (mL).

For cellulose content determination, the AOAC 973.18 method was followed. The treated sawdust was mixed with 72% sulfuric acid (15 mL) and placed in a water bath at 20–23 °C for 3 h. The sample was then repeatedly filtered and washed until the pH was neutral (7). After drying at 105 °C for 8 h, the cellulose content was calculated using the following Equation (4):(4)$$Cellulose\;content\left(\%\right)=\left[\frac{W_{1}-W_{2}}{W_{0}}\right]\times 100$$
where W_1_ = hemicellulose free sawdust weight; W_2_ = weight measured after H_2_SO_4_ addition; and W_0_ = weight after H_2_SO_4_ removal.

In compliance with TAPPI T222 om-22, the author determined the lignin percentage where sawdust, which was obtained through Equation (4), boiled with 72% sulfuric acid and entered into a crucible, and the rest of the drying and washing procedure remained the same as mentioned in earlier steps.

### 2.14. (H_2_O_2_ + Acetic Acid + KOH + TiO_2_ + Acrilonitrile) Treatment

Rahman et al. [72] worked on kempus-sawdust–unsaturated-polyester (UP) composites where the effect of chemically treated sawdust addition was discussed. The chemical treatment process was conducted in several steps, as outlined below:i.Wood Pulp Preparation (WP):A total of 200 g of sawdust was soaked in distilled water for 16 h. Following this, the sawdust was placed in a grinder vessel, with water added to fill 2/3 of the vessel, and stirred for 30 min. The mixture then underwent ultrasonication at 80 °C for 1 h in a sonicator bath, which helped remove excess water from the sawdust.
ii.Delignification of Wood Pulp (DWPF):For the delignification process, a bleaching solution of 500 g was prepared by mixing 248.75 g hydrogen peroxide (H_2_O_2_), 248.75 g acetic acid, and 2.5 g titanium oxide. This solution was boiled at 130 °C for 3 h in a fume hood. After boiling, the delignified wood pulp was washed with potassium hydroxide (KOH) and distilled water until the pH reached 7.
iii.Alkaline Treatment of DWPF:The delignified wood pulp (DWPF) was treated with a 6 wt.% KOH solution. This solution was made by dissolving 54 g KOH in 846 g of water, yielding 900 mL. Then, 60 g of DWPF was added, and the mixture underwent ultrasonication at 80 °C for further chemical modification.

## 3. Properties of Sawdust Composite

### 3.1. Sawdust Concrete (SDC)

The authors investigated the use of sawdust as a replacement for natural sand or cement in concrete mixtures, examining the effects of different compositions on the mechanical and physical properties of concrete. Specifically, they focused on the impact of adding treated sawdust fibers to the concrete mixture and compared the mechanical and durability characteristics enlisted in Table A1. In this section, the mechanical performance and durability test results of concrete mixtures containing treated sawdust are discussed. Key aspects considered include compressive strength, water absorption, the influence of curing time on the properties of the samples, and permeability.

#### 3.1.1. Compressive Strength of Treated SDC

Several studies have explored the effects of chemical treatments on the compressive strength and mechanical properties of sawdust–concrete composites. For instance, Siddique et al. [62], following the standard outlined in [73], assessed the compressive strength of treated sawdust concrete using cube-shaped specimens subjected to a compression load of 4.5 KN/s. For each sample, three specimens were tested, and the average values were calculated. Figure 4a shows the compressive strength of control concrete (BC) and sawdust-treated concrete (BW5 and BW10) at 7, 28, and 56 days. The control concrete (BC) has a consistently better strength throughout all curing durations, demonstrating the superior performance of conventional concrete without sawdust. In contrast, BW5 and BW10 combinations had reduced compressive strength after 7 days, indicating a slower strength development due to the addition of sawdust, which increases porosity and weakens the matrix. However, by 28 days, both BW5 and BW10 show considerable improvement, with BW5 approaching BC’s performance, highlighting the relevance of curing time in improving hydration and bond growth in sawdust-modified concrete. By 56 days, the difference between BC and BW5 narrows even more, indicating that prolonged curing can attenuate some of the strength loss associated with sawdust addition.

Figure 4b shows the influence of different sawdust percentages (0%, 5%, 10%, 15%, and 20%) on compressive strength after 7, 28, and 56 days. Mixtures containing 5% and 10% sawdust have the maximum compressive strength among sawdust-treated samples, especially after 28 and 56 days, indicating an ideal sawdust range for retaining structural integrity while increasing sustainability. However, as the sawdust percentage rises to 15% or 20%, compressive strength falls dramatically due to greater porosity and weaker interfacial bonding within the matrix. These findings highlight the vital necessity to keep sawdust content below 10% for structural applications while allowing enough curing time for optimum strength recovery.

Hewayde and Kubba [61] compared the compressive strength of treated sawdust concrete with that of a control concrete mixture, where sawdust was absent. The study found that concrete containing sawdust treated with distilled water and Ca(OH)_2_ had lower compressive strength compared to the control. However, when 3% CaCl_2_ was added to Ca(OH)_2_-treated sawdust, there was a sharp increase in compressive strength, surpassing the control mixture’s strength. Similarly, [59] investigated Ca(OH)_2_-treated sawdust and compared it with water-treated and Ca(OH)_2_ + varnish-coated sawdust. They found that chemical treatments, including water soaking, negatively impacted the compressive strength of the sawdust concrete. Untreated samples showed a strength of over 34 MPa, while treated sawdust significantly reduced this value. The addition of 10% varnish coating improved the strength slightly, but a 20% coating caused a significant drop. Furthermore, Dias et al. [63] conducted a comparative study between three different types of chemically treated wood waste-derived cement pastes (WW1, WW2, WW3) and a control cement paste. The chemical treatment was identical for each type of wood waste.

They found that the control cement paste had the highest compressive strength, while chemical treatments harmed the wood waste-based cement pastes. Moreover, increasing fiber content from 5% to 10% or 15% further reduced the strength. The size of the wood waste (10 mm vs. 4 mm) did not significantly affect compressive strength, as both sizes produced similar results.

In another study, [38] explored the combination of furnace slag as a replacement for Portland cement and sawdust as a replacement for fine aggregate. They used NaOH-treated sawdust with untreated blast furnace slag in different compositions (10%, 20%, and 30% slag combined with 5%, 10%, and 15% sawdust). Their findings revealed that a mixture of 10% ground granulated blast furnace slag (GGBS) and 5% sawdust achieved the highest compressive strength, showing a 33.34% improvement over the control mixture (30 MPa). However, higher slag and sawdust content led to a gradual decline in strength, with the 30% slag and 15% sawdust combination providing only half the strength of the control. Hassen and Hameed [64] focused on sawdust as a partial replacement of the total dry content (ranging from 10% to 50%) in concrete. They examined three types of sawdust: untreated, NaOCl-treated, and NaOCl-treated + washed. Using the ASTM/C 109 standard for compression tests, they found that chemical treatments generally decreased the compressive strength. However, sawdust that was treated and then washed showed the highest strength. Untreated sawdust mixtures performed better than treated sawdust, but NaOCl-treated and washed sawdust at 15% replacement offered the greatest compressive strength. Higher sawdust content, particularly 50%, resulted in the lowest strength, while 10% sawdust showed the highest strength. Finally, [67] conducted compatibility tests on sawdust–cement composites using sawdust from three different wood species (*Triplochiton scleroxylon*, *Entandrophragma cylindricum*, and *Klainedoxa gabonensis*), treated with a mixture of ethanol, NaCl, and H_2_SO_4_. Following the BS EN 310:1993 standard, they found that *T. scleroxylon*-derived-sawdust composites had the highest modulus of elasticity (696.1 N/m²) due to better bonding with cement. *E. cylindricum* showed a lower modulus (625.9 N/m²), while *K. gabonensis* had poor compatibility with cement, breaking during manufacturing.

A similar study by [74] evaluated the modulus of rupture (MOR) and shear strength of sawdust-cement composites made from the same wood species. *T. scleroxylon* again outperformed the others due to its lighter weight, leading to improved bonding with cement. The study also determined that hot water treatment was the most effective method for removing inhibitory substances from sawdust, resulting in higher shear strength and MOR compared to cold water or ethanol treatments. Hot-water-treated-sawdust composites exhibited better overall compatibility with cement. Kong et al. [56] conducted experiments on red-mud-sawdust-geopolymer composites, where sawdust was treated with a solution containing 5% NaOH, 20% H_2_O_2_, and KOH. The treated sawdust was mixed with red mud in varying proportions, from 10% to 60% in 10% intervals. The compressive strength decreased progressively with increasing sawdust content after the initial 10% sawdust addition, reaching the lowest value at 60% sawdust.

Saeed et al. [60] highlighted the benefits of chemical treatments by applying four different treatments to sawdust, categorized into groups 1, 2, 3, and 4. The sawdust was used as a fine aggregate in cement mixtures, ranging from 0% to 35% at 5% intervals. However, all groups showed a significant decline in compressive strength with increasing fiber content. Group 4, where the sawdust was treated with Ca(OH)_2_ and varnish, exhibited the highest compressive strength at 33.4 MPa for 5% sawdust addition. In contrast, group 1, with untreated sawdust, achieved only 22 MPa for the same 5% addition, followed by a substantial decrease at higher sawdust levels.

#### 3.1.2. Durability of Treated SDC

Curing time significantly influences the performance of concrete structures. For both CA and CB compositions, concrete cured for 56 days exhibited better performance compared to concrete cured for 7 and 28 days, regardless of the chemical treatment applied. The difference between 7 and 28 days of curing was notable. For instance, 5% water-treated sawdust concrete (SDC) showed a compressive strength of 19 MPa after 7 days of curing, which increased by 42% to 27 MPa after 28 days. However, after 56 days of curing, the increase was only 3.45%. A similar pattern was observed in Na_2_SiO_3_-treated SDC. For example, 5% Na_2_SiO_3_-treated SDC with CB composition exhibited a 47.36% increase in compressive strength after 28 days and a smaller 5.35% increase after 56 days.

Additionally, the authors measured water absorption and penetration depth in water-treated SDC, considering the percentage of sawdust added. The capillary water absorption of water-treated sawdust concrete increased sharply with increasing sawdust content. The 20% treated sawdust had the highest water absorption value of 5 mm, with a penetration depth of 4.2 mm. Although water absorption continued to rise to 20% with sawdust addition, penetration depths for 10% and 15% sawdust were approximately 2 mm and 3.1 mm, respectively, showing a gradual slope in the graph [62]. A similar durability trend was observed in the study by [61], where they examined the compressive strength of various chemically treated sawdust concrete (SDC) samples under identical curing times. Echoing the findings from [62], the compressive strength of all chemically treated SDC samples increased after 28 days of curing compared to those cured for 7 days. Specifically, control concrete, distilled water-treated SDC, Ca(OH)_2_-treated SDC, and Ca(OH)_2_ + CaCl_2_-treated SDC showed compressive strength increments of approximately 45%, 60%, 70%, and 36.36%, respectively, after 28 days compared to their 7-day cured counterparts. In this paper, the authors did not conduct any water uptake or chemical penetration tests to support their claims regarding the durability of concrete after each chemical treatment. Similarly, Dias et al. [63] emphasized the importance of proper curing for concrete structures, noting that 28-day cured wood samples showed better compressive strength than 7-day cured samples. Specifically, they observed strength increases of 21.5%, 14.7%, and 2.5% for WW1, WW2, and WW3 samples, starting from initial values of 34.8 MPa, 28.9 MPa, and 35.7 MPa, respectively.

Conversely, [59] used the same sawdust across different chemical treatments to assess their effects on durability over 7, 14, and 28-day curing periods. Untreated SDC had compressive strength values of 34.5 MPa, 37.03 MPa, and 38.96 MPa after 7, 14, and 28 days of curing. Among the six treatment types, sawdust treated with Ca(OH)_2_ and a 10% coating yielded comparable results to untreated SDC, with compressive strengths of 15.13 MPa, 11.3 MPa, and 18.1 MPa after 7, 14, and 28 days, respectively. On the other hand, water-soaked SDC with 20% fine aggregate replacement showed the lowest compressive strength results, with 2.08 MPa, 3.9 MPa, and 6.3 MPa for 7, 14, and 28 days of curing. A significant decline in strength was noted when 20% sawdust was added instead of 10%, as seen in Table 2. The study reveals that the compressive strength of sawdust varies depending on the treatment and replacement levels. Untreated sawdust maintains the highest strength, while unsoaked replacements show reduced strength due to poor bonding with the cement matrix. Soaking sawdust reduces hydrophilicity, allowing better particle-matrix adhesion. Calcium hydroxide treatment modifies the sawdust’s hydroxyl groups, enhancing matrix compatibility. Treatment sawdust improves bonding by 28 days, with lower replacement levels performing more consistently. This suggests that chemical treatments can enhance sawdust’s sustainability as a composite material.

A similar analysis is presented in [46], indicating that 28-day cured sawdust concrete exhibited higher compressive strength compared to 7-day cured samples in all cases. For mixtures combining 20% blast furnace slag with 5–15% treated sawdust, there was a significant difference in compressive strength between the 28-day and 7-day cured samples, whereas in other cases, this difference was minimal. Additionally, the addition of sawdust and ground granulated blast furnace slag (GGBS) reduced water absorption. The control concrete mixture had a water absorption rate of 2.5%, but all variations in blast furnace and sawdust combined SDC showed lower water absorption. The mixture of 10% blast furnace and 5% sawdust exhibited the lowest absorption rate at 1.1%. But sawdust addition beyond the optimal range can create a negative effect on compressive strength, as shown in Figure 5a. Figure 5b shows water absorption in untreated and treated sawdust composites with different sawdust percentages (10%, 20%, 30%, 40%, and 50%). Water absorption rises with increased sawdust concentration, indicating sawdust’s intrinsic porosity and hydrophilicity. However, treated sawdust consistently absorbs less water than untreated sawdust at all percentages, confirming the efficacy of chemical treatments in lowering hydrophilicity and increasing water resistance. At lower sawdust percentages (10% and 20%), the difference in water absorption between treated and untreated sawdust is less noticeable, but the gap grows dramatically at higher percentages (40% and 50%). This suggests that chemical treatment becomes more important as sawdust content increases, mitigating the negative effects of increased permeability and improving durability. Overall, the findings indicate that chemical therapy and optimum sawdust levels are critical to balancing water resistance and strength in sawdust [54].

From Boasiako et al. [67], *T. scleroxylon* absorbed 8.82% moisture, which was the lowest compared to 9.52% for *E. cylindricum* sawdust-cement blocks. Additionally, *T. scleroxylon* exhibited better density than *E. cylindricum*, with values of 630.97 kg/m³ and 572.98 kg/m³, respectively. This observation is supported by [74], finding that *T. scleroxylon* showed better performance than *E. cylindricum* with all types of chemical treatments.

### 3.2. Sawdust-Polymer Composite (SDPC)

Different types of chemical treatment and their impact on SDPC, especially tensile strength and morphological investigations, are characterized in Table A2.

#### 3.2.1. Tensile Properties

Hossain et al. [47] prepared a 114 mm × 7 mm × 6 mm dog-bone-shaped specimen according to the ASTM D3039 standard, where a 2 mm/min crosshead speed was applied to obtain the results. From their observation, 10% NaOH treatment has a positive impact on increasing the tensile strength in sawdust-polyester composites where Garjon (*Dipterocarpus tubinatus grtn*) sawdust exhibited the greatest tensile strength, either 10% addition or 5% addition, compared with Gamari (*Gmelina arborea*)’s lowest tensile strength. In each case, treated SDPC showed better tensile strength than untreated SDPC, albeit fiber addition caused lower tensile strength. For example, Garjon sawdust composites had 34 MPa with 5% sawdust addition, but it reduced to 31 MPa after 10% loading. Similarly, Kerosene shifted from 33 MPa to 31 MPa, and Gamari from 32 MPa to 28 MPa as well. Consequently, Haque et al. [48] used a 4% NaOH solution for *Swietenia mahagoni* sawdust treatment, and the tensile test was carried out following the same standard and considerations except for crosshead speed. They observed that there was a 36.7% increment in tensile strength after chemical treatment, whereas untreated sawdust composites provided only 15.83 MPa. Similarly, 4% NaOH-treated SDPC presented a 1545.2 MPa tensile modulus with a 1230.7 MPa modulus of the untreated sample.

After that, Ganesan et al. [70] also followed the ASTM D3039 standard for teak sawdust, but specimens were 250 mm × 25.4 mm dimensioned flat sections. After the application of 2 mm/min crosshead speed, authors observed that 14% tensile strength increased due to chemical treatment by 5% NaOH on sawdust compared with untreated samples. But [51] made their tensile specimen into 105 mm × 7.5 mm × 4.5 mm dimensions, though the standard remained the same. They observed that the tensile strength of ABS-sawdust composites declined drastically with up to 10% sawdust addition and then rose a little bit at 15% before another decrement at 20% addition. Again, the elastic modulus of sawdust increased slightly for 5% sawdust; after that, a sharp increasing trend up to 15% of sawdust was observed with a figure of 825 MPa (Figure 6a). Completely reverse picturesque as seen in Figure 6b, where after 5% sawdust addition, a gradual decreasing phenomenon was exhibited by treated sawdust composites with the lowest value of 4.65% approximately. ABS–sawdust composites reach their maximal elastic modulus at 15% sawdust addition, suggesting a balance of filler quantity and matrix integrity. However, increasing sawdust concentration reduces elongation, showing the filler’s stiffness. Treated composites and hybrid flame retardant systems increase interfacial characteristics but suffer from overloading difficulties, highlighting the need for stiffness and ductility tuning. In this paper, the authors concentrated more on the fiber loading effect on SDPC rather than a comparative study between treated and untreated sawdust, which was conspicuously analyzed by Ganesan et al. [70].

Akter et al. [57] investigated HDPE-sawdust-polymer composites, where sawdust was added in increments of 5 wt.% up to 45 wt.%. They compared untreated sawdust with chemically treated sawdust and analyzed the effect of fiber loading on hybrid composites. Composites with benzoyl-peroxide-treated sawdust exhibited greater strength than untreated ones. A notable difference was observed with 5 wt.% sawdust in hybrid composites, but at 25 wt.% addition, the strength of the treated sawdust composite was lower than that of the untreated one. As more sawdust was added, the strength difference between treated and untreated composites became less significant. A similar trend was seen in elongation measurements—up to 15 wt.% sawdust, the difference between treated and untreated samples was significant, but afterward, the elongation of treated sawdust composites decreased and became lower than that of untreated composites, reaching the lowest value of 2% at 45 wt.% addition.

Prosper et al. [52] followed the ASTM D638 standard to determine the tensile strength and Young’s modulus of sawdust-oil-bean-pod-shell-epoxy hybrid composites, using 10% NaOH for chemical treatment. Both tensile strength and Young’s modulus showed an increasing trend with 10%, 20%, 30%, 40%, and 50% fiber loading. However, beyond this, the tensile properties started to decline. Benjamin et al. [54] performed similar research, adding sawdust fibers from 10% to 30% in increments of 5% after a 5% NaOH chemical treatment. They observed a steady increase, with tensile strength reaching 28 MPa, Young’s modulus hitting 1200 MPa, and tensile energy reaching approximately 12 N∙m at 30% fiber loading. However, no morphological analysis was conducted on the sawdust–OBPD-epoxy hybrid composites.

The positive impact of adhesion-enhancing components on the mechanical properties of chemically treated sawdust was demonstrated by Shulga et al. **[56]**. They followed the ASTM D638 tensile standard, applying 50 mm/min crosshead velocity using a 0.5 KN UTM machine on LPEC-enhanced, ammoxidized, and hydrolyzed aspen–sawdust composites. The tensile strength of untreated sawdust was 26.2 MPa, which increased to 38.5 MPa and 39.8 MPa after treatment with (HCl + LPEC) and (NH_4_OH + LPEC) solutions, respectively. A similar trend was noted in tensile modulus analysis, with untreated sawdust displaying 725 MPa, which rose to 1028 MPa and 1120 MPa after treatment.

Kusuktham et al. [71] evaluated the tensile properties of rubber-wood–HDPE hybrid composites according to the ISO 527-2 standard. Dumbbell-shaped specimens measuring 105 mm × 25 mm × 3 mm were tested using a 5 mm/min crosshead speed. In this study, HDPE acted as the matrix, and CaCO_3_ and sawdust treated with vinyltriethoxysilane (VTES) were used as fillers. They compared HDPE–sawdust and HDPE–CaCO_3_ composites, adding 2.5%, 5%, 7.5%, and 10% sawdust or CaCO_3_, individually treated with VTES. VTES-treated–sawdust composites exhibited better mechanical properties than raw sawdust composites, with 10% sawdust providing the highest tensile strength and 2.5% the lowest. Similarly, HDPE–CaCO_3_ composites showed the highest tensile strength at 7.5% addition. However, they noted that VTES had only a minimal effect on improving mechanical properties. Rahman et al. [68] studied the use of 5% KOH-treated delignified kempas sawdust as a filler material in sawdust–unsaturated-polyester composites (WFUPC) and conducted a comparative analysis with cyanoethyl-cellulose-fiber-oriented–unsaturated-polyester composites (CEFUPC) and cellulose-fiber-unsaturated-polyester-resin composites (CFUPC). Their findings showed that WFUPC had superior mechanical properties, such as tensile strength and Young’s modulus, with fiber loading up to 15 wt.%, after which there was a decline. Among the composites, CEFUPC exhibited the highest mechanical properties and better interfacial adhesion compared to wood pulp and cellulose fiber composites. In another paper published the same year, Rahman et al. [55] obtained similar results from delignified sawdust composites, although the scope was slightly different as they only compared delignified sawdust composites with raw sawdust composites, unlike the previous study, which included CEFUPC and CFUPC. Fiber addition positively impacted tensile strength and modulus up to 15%, after which the properties began to decline. Another related study by [46] used 5% NaOH treatment on sawdust to fabricate sawdust-polypropylene composites, following the same fiber addition sequence. The trend in tensile strength and Young’s modulus was consistent, with both raw and treated sawdust composites showing improvement up to 15% fiber loading, followed by a sharp decline. In all cases, treated sawdust composites outperformed raw sawdust ones. Similarly, Ferede [75] conducted research using the same parameters as Rahman et al. [46] but varied the fiber addition sequence, adding 10% to 50% sawdust in 10% increments. He observed an increasing trend in the mechanical properties of sawdust-PP composites up to 40% fiber loading, after which the properties declined significantly. Unlike [55], who analyzed the effect of fiber addition using SEM and FTIR, [75] only focused on mechanical properties and water absorption tests.

The flame retardant loading condition investigated by Suoware et al. [53] demonstrates the trade-offs between improving tensile strength and maintaining tensile modulus in NaOH-treated-sawdust composites. The addition of 12% ATH significantly enhances the tensile strength by 84% compared to composites without flame retardants. This improvement is attributed to the superior interfacial adhesion between ATH and the polyester resin, which facilitates effective stress transfer within the composite matrix. However, other flame retardant loadings, such as 12% APP-GAP and the hybrid 18% ATH/APP-GAP, result in lower tensile strengths (12.4 MPa and 16.5 MPa, respectively). These differences arise due to variations in the interaction of each flame retardant with the polyester matrix and sawdust, with ATH exhibiting better compatibility.

On the downside, all flame retardants negatively affect the tensile modulus, with composites containing 12% ATH, 12% APP-GAP, and 18% ATH/APP-GAP exhibiting lower values (1.50 MPa, 1.20 MPa, and 1.32 MPa, respectively) compared to the untreated composite’s 1.55 MPa. This reduction likely stems from the flame retardants acting as stress concentrators within the matrix, impeding the modulus despite enhancing strength in specific cases. These findings highlight that while ATH improves tensile strength due to its compatibility, the overall balance between strength and stiffness must be carefully considered in flame-retardant-sawdust composites, as reflected in Figure 7.

#### 3.2.2. Morphological Observation

Chemically treated sawdust-polyester composites (SDPC) exhibited improved tensile strength due to enhanced interfacial adhesion between the unsaturated polyester (UP) matrix and the sawdust fillers. This improvement is attributed to the chemical treatment, which reduced the size of the filler material, thereby enhancing the aspect ratio and promoting better adhesion with the UP resin. Furthermore, the NaOH treatment removed lignin from the sawdust fibers, increasing the adhesive surface area of the fibers. However, the addition of sawdust fibers led to lower mechanical properties, primarily due to weak bonding in the composites, as observed in the SEM images, where sawdust clustered at various points. Additionally, fiber agglomeration, debris presence, fiber debonding, and air voids were identified as significant factors contributing to the sharp decrease in tensile strength [76]. Surface integrity can also be improved through chemical treatment, as evident in the [48] study. One key aspect of their study is that they did not examine the fractured surfaces after mechanical testing; instead, they focused only on the fabricated surfaces. Figure 8a shows the untreated sawdust with visible voids, while the chemically treated sawdust in Figure 8d reveals a smooth, continuous surface, which enhances the interfacial bonding with the matrix phase (unsaturated polyester resin). They also analyzed the treated SDPC using FTIR spectroscopy and observed that at the 1726 cm^−1^ wavelength, corresponding to the resin, the same C=O peak for the treated sawdust was less prominent at 1710 cm^−1^. This suggests that the 26 cm^−1^ redshift is due to the formation of hydrogen bonds between hydroxyl and carbonyl groups. The C=O bond peak became less distinct in the treated sawdust composite because the removal of lignin from the sawdust contributed to the formation of a hydrophobic composite with improved tensile strength. Additionally, Neher et al. [51] investigated how frequency varies with different fiber loadings, as depicted in Figure 9, with a frequency range of 650 to 4000 cm^−1^. In the absence of sawdust (0% loading), the ABS polymer matrix exhibited multiple C-H bonds. However, with the addition of fibers, these bonds were disrupted, and a new C≡C bond emerged after 20% sawdust addition. The impact of this change is illustrated in Figure 6a,b, which show the elastic modulus and elongation values for varying fiber percentages.

Ganesan et al. [70] examined the impact of a cryogenic environment on NaOH-treated SDPC. They conducted cryogenic treatments for 30, 60, and 90 min and presented the SEM findings in Figure 10. Their results showed that 60 min of treatment provided better mechanical performance compared to the 30 and 90 min treatments. The cryogenic treatment introduced significant internal stress in the SPDC, which weakened the interfacial bonding between the matrix and filler materials. This internal stress increased with longer treatment times, accelerating material failure as the fiber–matrix bonding deteriorated. In Figure 10a, voids are visible after 30 min of cryogenic treatment, though no cracks formed. At 60 min, fibers exhibited strong bonding with the matrix. However, the continued buildup of internal stress after 90 min led to fiber fibrillation, disrupting the stress distribution and reducing the material’s strength, as seen in Figure 10d.

Akter et al. [57] performed both SEM analysis and FTIR spectroscopy and concluded that sawdust treated with benzoyl peroxide demonstrated superior interfacial adhesion and compatibility with the matrix phase. In contrast, untreated sawdust exhibited frequent fiber pullouts on the fiber surfaces, weakening its adhesion to the matrix. Furthermore, the FTIR study of NaOH-treated-sawdust composites demonstrates significant structural changes owing to benzoyl peroxide treatment. A decrease in the O-H signal at 3400 cm^−1^ shows the degradation of hydroxyl groups, which is necessary for turning cellulose into dialdehyde cellulose. The presence of C=O peaks at 1000–1200 cm^−1^ indicates the creation of aldehyde groups from the oxidation of cellulose’s anhydroglucose rings at carbon 2 and carbon 3. The stable fundamental structure of cellulose is shown by persistent C-H peaks about 2900 cm^−1^. However, the existence of C=N peaks at 1600–1650 cm^−1^ shows probable cross-linking or interactions within the composite. These alterations improve interfacial bonding in the composite, making the modified cellulose more compatible with the polymer matrix.

Shulga et al. [66] analyzed the effect of LPEC on sawdust fibers through SEM, focusing on the surface roughness of untreated sawdust. The uneven surface contributed to the formation of microcracks and defects under applied loads. SEM images highlight fiber pullout, debonding, and air voids that accelerate poor adhesion. In contrast, it is evident that improved surface quality with less fiber pullout. Kusuktham et al. [71] studied the effect of VTES on sawdust surfaces, unmodified sawdust with high surface roughness, and long needle-like structures. Meanwhile, the smoother surface of sawdust, treated with VTES, was coated with a silane layer.

Rahman et al. [72] examined how delignification affected the properties of sawdust composites, using SEM and FTIR to reveal internal morphological changes under load. They found large, non-uniform pores in the untreated sample (Figure 10e). After delignification, a reduction in pore size results in a smoother surface due to the removal of the lignin and waxy contents (Figure 10f). Unsaturated polyester (UP) addition helps to reduce pores significantly in raw sawdust, as shown in Figure 10g. Still, there are tiny gaps between the filler and the matrix. This suggests that the polymer-filler interaction was weak, making the filler less compatible with the UP matrix. In contrast, the DWS-UP-C demonstrated improved adherence between filler and matrix. This is owing to the strong interfacial bonding between the filler and matrix, which allowed for efficient stress transmission from the matrix to the filler in the composite. A smooth surface with better interfacial adhesion between the treated sawdust and the UP resin enables efficient stress transfer (Figure 10h). FTIR analysis revealed a large peak at 3300 cm^−1^ in untreated sawdust, indicating water absorption due to hydroxyl groups, which was significantly reduced after delignification. Additionally, Rahman et al. [46] explored the effect of fiber loading on sawdust-polypropylene composites. They noted that the presence of microvoids, fiber pullout, and fiber agglomeration hindered efficient stress transfer between the fibers and the matrix, weakening interfacial bonding. This bond degradation worsened with increased fiber volume.

## 4. Conclusions

The pervasive acceptance of sawdust has grown due to promising alternatives in concrete and polymer composite fabrication. In this regard, extensive research is carried out in SDC and SPDC focusing on the circular economy approach, where chemical treatment has become a proven technique to draw significant mechanical and physical properties. In the case of SDC, a wide range of treatments, including NaOH, Ca(OH)_2_, and Na_2_SiO_3_, enhances the characteristics, especially compressive strength and durability. However, appropriate sawdust selection and percentage of fiber balance are crucial, as excessive addition leads to weakly bonded composites. Na_2_SiO_3_ exhibited superior results compared with untreated and water treatment. Although it has improved, it still lags behind the performance of control concrete mixtures. Moreover, adequate curing time in compressive strength results is paramount. Multiple studies in this article demonstrated the 28- and 56-day curing schemes have a significant impact on compressive strength compared to the 7 days. In terms of the durability test of SDC, water uptake tests were conducted in almost every research study and a similar observation was concluded. Irrespective of treatment, sawdust addition promotes permeability, where untreated sawdust exerted the highest water absorption phenomena. Though Ca(OH)_2_ and varnish are considered an effective way to alleviate this issue, lower mechanical strength from this technique impedes their wider application. Therefore, achieving both strength and durability in SDC requires carefully optimizing the balance between chemical treatments, fiber additions, and curing methods.

For sawdust-polymer composites (SDPC), the tensile properties were greatly improved with chemical treatments like NaOH, benzoyl peroxide, and VTES. Removing lignin and enhancing interfacial bonding between the matrix and sawdust fibers were key contributors to this improvement. Compared with untreated sawdust, NaOH-treated sawdust illustrated a rapid surge in tensile strength and elastic modulus results. From SEM and FTIR analysis, higher surface integrity, such as a smooth surface and fewer defects, proliferate the effective load distribution that results in greater mechanical stability. However, likewise SDC, excessive fiber addition can cause fiber agglomeration and poor interfacial adhesion, eventually leading to lower tensile strength. Morphological investigations revealed that chemical treatment is beneficial in reducing the voids, surface roughness, and fiber pullout significantly more than those frequently observed in untreated sawdust.

Overall, NaOH treatment is the most effective method for sawdust composites because it removes contaminants and improves interfacial interaction between sawdust and matrix components. However, over-treatment can damage cellulose, lowering its mechanical qualities. Although Ca(OH)_2_ treatment improves water resistance, it generally results in poor mechanical performance due to inadequate chemical alteration and fiber–matrix interaction. Surface coatings such as varnish can minimize water absorption while compromising mechanical strength. Hybrid treatments, such as NaOH mixed with coupling agents like VTES or benzoyl peroxide, address both interfacial bonding and moisture resistance. However, these treatments necessitate precise optimization to avoid problems such as fiber agglomeration and cost inefficiency. In summary, this review highlights sawdust’s potential as a sustainable and adaptable material for composite applications while stressing the importance of carefully managing treatment parameters and fiber content. Future studies should aim to refine treatment techniques to improve the mechanical properties of sawdust composites while ensuring environmental sustainability. Advancing hybrid treatment approaches and investigating new chemical agents could further expand sawdust’s contribution to solving global material shortages, especially in the construction and manufacturing sectors.

## 5. Future Scope

The study’s future goals include improving the performance and application possibilities of sawdust composites. Future study might focus on creating enhanced surface treatments or bio-based changes to improve sawdust’s compatibility with polymer and cement matrices, hence boosting mechanical and thermal performance. Hybrid composites including sawdust and additives such as nanoclays or graphene may have higher stiffness, flame retardancy, and durability. Furthermore, improving manufacturing procedures like extrusion or 3D printing may solve concerns such as particle aggregation while ensuring consistent material qualities. Long-term durability tests under a variety of environmental circumstances are required to assess aging behavior and biodegradability, assuring the composites’ dependability in real-world applications. Furthermore, economic and environmental studies, such as life-cycle cost and carbon footprint analysis, can help to confirm sawdust composites’ sustainability.

## Figures and Tables

**Figure 1 polymers-16-03289-f001:**
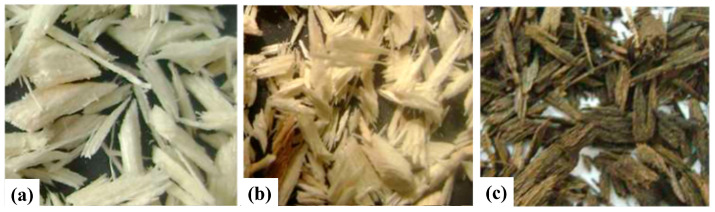
For mechanical performance evaluation, sawdust is grouped into 4 stages including (**a**) untreated sawdust; (**b**) Ca(OH)_2_ washed sawdust; and (**c**) treated with cutback asphalt; [60] © 2020 Eyasu. Licensee Hindawi. Used under CC-BY 4.0.

**Figure 2 polymers-16-03289-f002:**
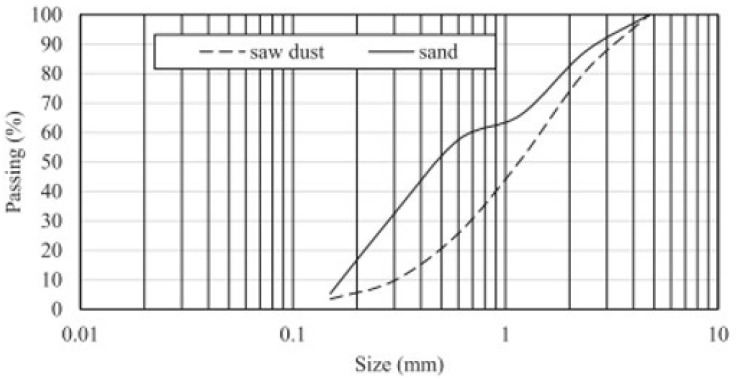
Sawdust particle size distribution information [62] © 2024 Elsevier. Used with permission.

**Figure 3 polymers-16-03289-f003:**
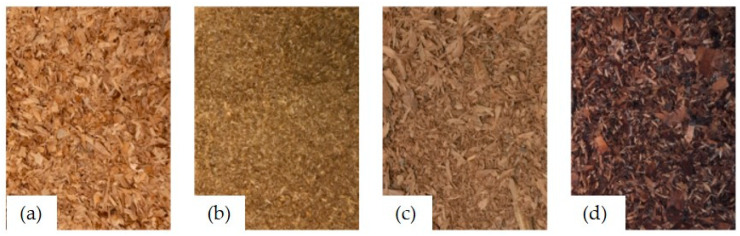
Four different types of sawdust samples sourced from (**a**) sawmill; (**b**) and (**c**) agricultural post and fence (differently sourced); and (**d**) railway sleepers before chemical treatment [63] © 2021 Authors. Licensee Elsevier Ltd. Used under CC BY-NC-ND.

**Figure 4 polymers-16-03289-f004:**
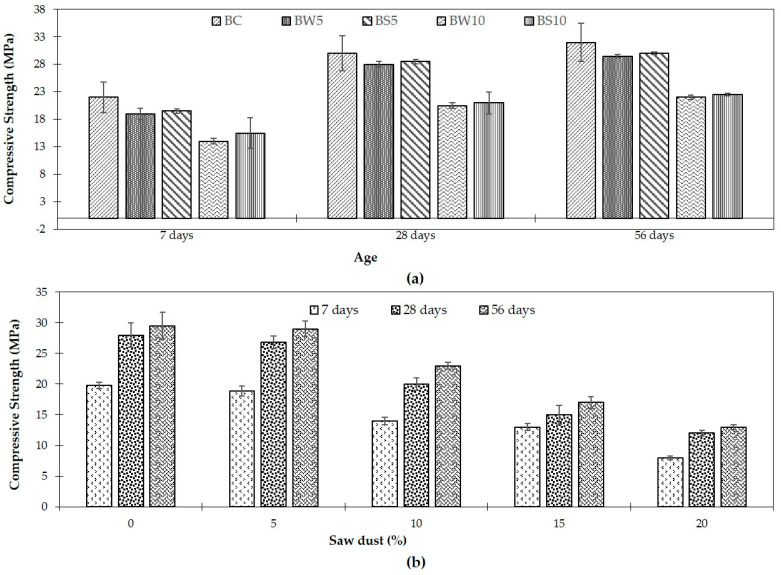
(**a**) Compressive strength for both water-treated and Na_2_SiO_3_-treated SDC following BC composition [62] © 2024 Elsevier. Used with permission. (**b**) Water-treated SDC compressive properties in CA-concrete composition [62] © 2024 Elsevier. Used with permission.

**Figure 5 polymers-16-03289-f005:**
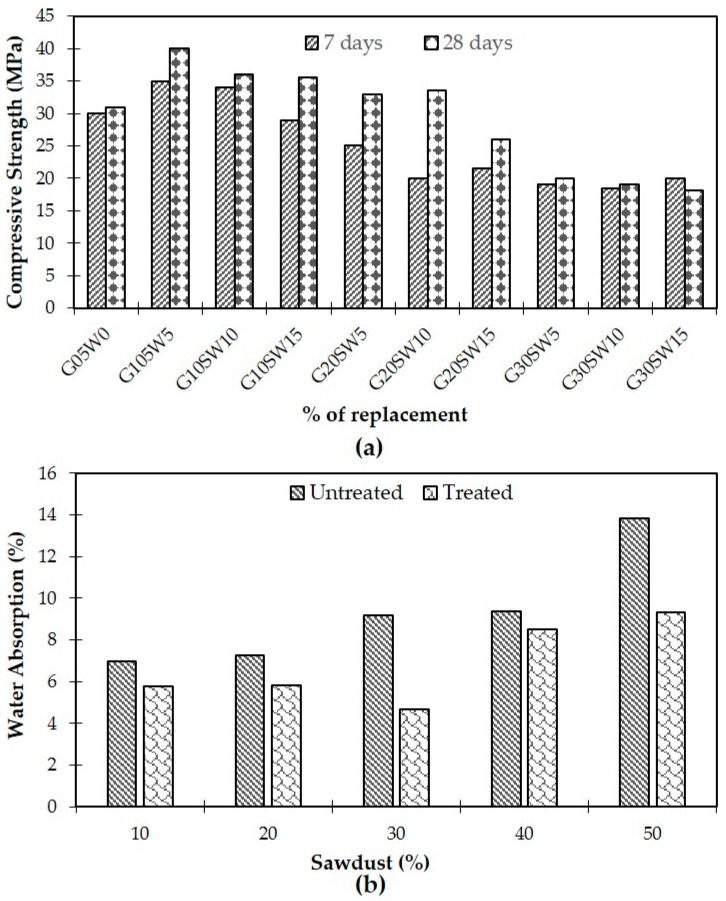
Curing effect on (**a**) compressive strength [46] © 2020 Elsevier Ltd. Used with permission; (**b**) water absorption [64] © 2020 Authors. Licensee IOP Publishing Ltd. Used under CC-BY. 3.0.

**Figure 6 polymers-16-03289-f006:**
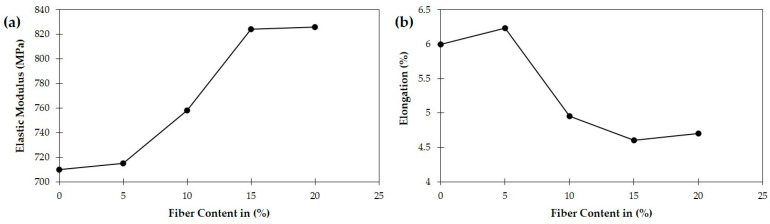
(**a**) Elastic modulus and (**b**) elongation of ABS-sawdust composites [51] © 2020 Authors. Licensee Scientific Research Publishing Inc. Used under CC-BY 4.0.

**Figure 7 polymers-16-03289-f007:**
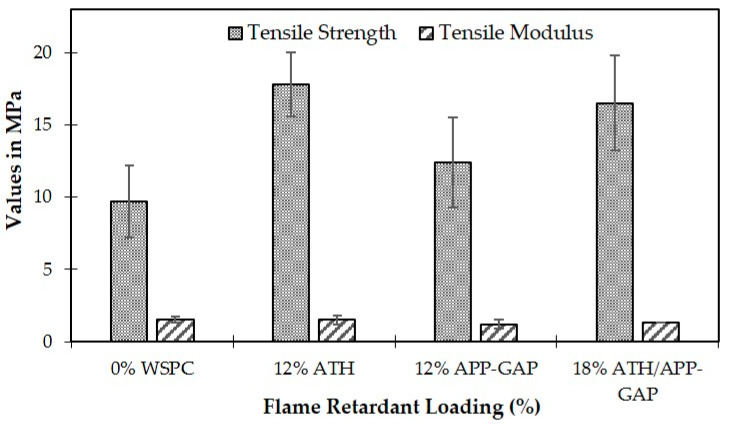
Mechanical properties (tensile strength and tensile modulus) with response to different flame retardant compositions [53] © 2019 John Wiley & Sons, Ltd. Used with permission.

**Figure 8 polymers-16-03289-f008:**
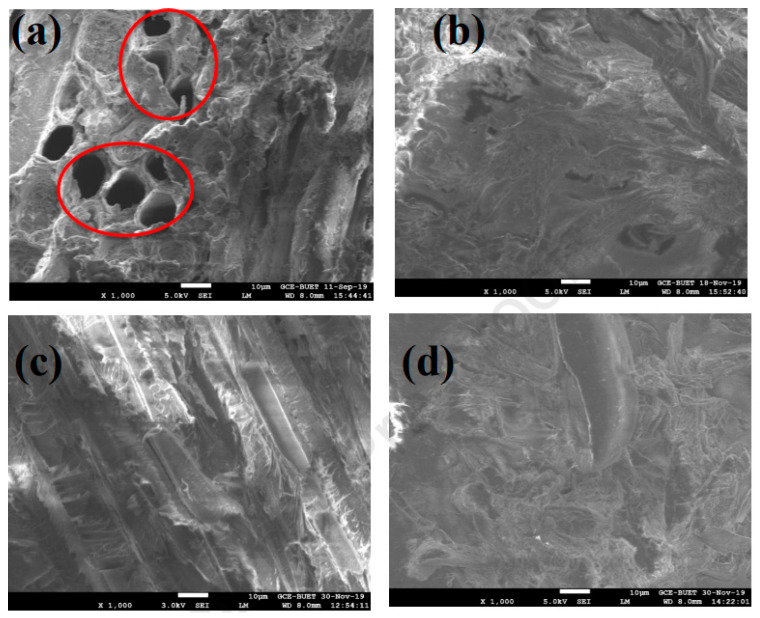
SEM photograph of (**a**) untreated sawdust; (**b**) treated sawdust; (**c**) untreated SDPC; and (**d**) treated SDPC [48] © 2020 Elsevier Ltd. Used under CC BY-NC-ND.

**Figure 9 polymers-16-03289-f009:**
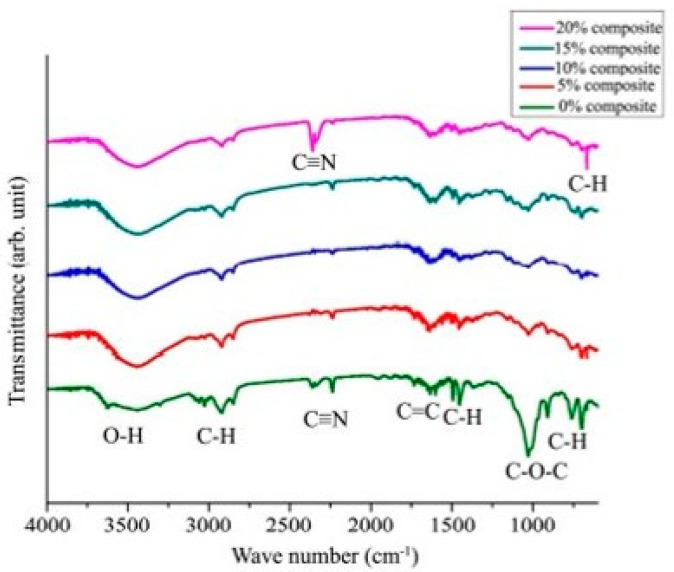
FTIR analysis for different volumes of sawdust addition [51] © 2020 Authors. Licensee Scientific Research Publishing Inc. Used under CC-BY 4.0.

**Figure 10 polymers-16-03289-f010:**
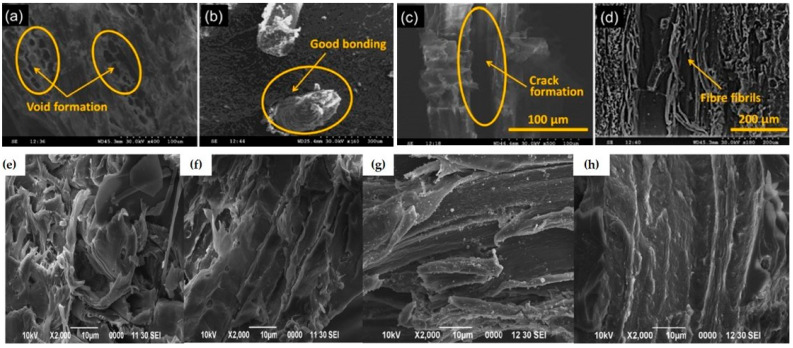
SEM photograph of SPDC under cryogenic treatment of (**a**) 30 min; (**b**) 60 min; and (**c**) 90 min. (**d**) Fibril formations [70] © 2021 Authors. Licensee MDPI. Used under CC-BY 4.0. (**e**) Pores in raw sawdust; (**f**) pores in delignated sawdust; (**g**) UP-raw-sawdust-smoothed surface; and (**h**) UP-delignated-sawdust-smoothed surface [72] **©** 2021 society of plastic Engineers. Used with permission.

**Table 1 polymers-16-03289-t001:** Sawdust chemical composition [46] © 2024 Elsevier. Used with permission.

Sample No.	Chemical Property	Results
1	MgO	7.09
2	CaO	40.67
3	${Al}_{2}O_{3}$	13.69
4	${SiO}_{2}$	34.58
5	${Fe}_{2}O_{3}$	0.44
6	${SO}_{3}$	0.56
7	$K_{2}O$	0.32
8	Cl	-
9	${Na}_{2}O$	0.15

**Table 2 polymers-16-03289-t002:** Different chemical-treated sawdust concrete (SDC) compressive strengths [59] © 2020 Authors. Licensee CIC, Used under CC-BY 4.0.

Scenario	Strength (MPa)		
	7 Days	14 Days	28 Days
Untreated Sawdust	34.5	37.03	38.96
Unsoaked sawdust with 10% replacement	8.77	9.31	9.8
Unsoaked sawdust with 20% replacement	2.08	3.9	6.3
Soaked sawdust with 10% replacement	16	15.9	18.07
Soaked sawdust with 20% replacement	11.5	13.8	17.8
Sawdust treated with calcium hydroxide + 10% MC	15.13	11.3	18.1
Sawdust treated with calcium hydroxide + 20% MC	3.24	5.5	12.3

## Data Availability

The data presented in this study is available in the manuscript. For further enquiries, please contact the Corresponding author (mohammad.khondoker@uregina.ca).

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
