# Peer review of "Effect of Treatment Methods on Material Properties and Performance of Sawdust-Concrete and Sawdust-Polymer Composites"

_polymers, 2024, doi:10.3390/polym16233289_

Round 1
Reviewer 1 Report
Comments and Suggestions for Authors
1. Technically discuss the scenario of Table 2, why there is an increase or decrease in the strength?
2. Is the specimen prepared for varying 10 to 50%? If so provide the inference for it as per the figure 5b.
3. Provide the novelty of this review.
4. Author can include future scope.
5. Author can include the summary either in the way of figure or table to explain the inference.
6. Discuss in detail about the elastic modulus and elongation of ABS-sawdust in detail, can compare it other variants discussed.
7. Provide reason for the flame retardant loading condition as shown in figure 7.
8. Effect of treatment as shown in the SEM images 8 and 10, can be elaborated for the clarity.
9. FTIR shown in explained technically.
Comments on the Quality of English LanguageEnglish should be improved for the clarity
Author Response
1. Technically discuss the scenario of Table 2, why there is an increase or decrease in the strength?
Response: Thanks for your advice. A technical discussion is added there with proper explanation.
2. Is the specimen prepared for varying 10 to 50%? If so provide the inference for it as per the figure 5b.
Response: Saw dust addition increases up to 50% addition with a 10 % increment rate. It is observed that sawdust addition after the optimal range can deteriorate the overall performance. It will follow the same decreasing trend even after 50% sawdust addition. Untreated sawdust is more hydrophilic and chemical treatment reduces the water content from sawdust. But the water washing scheme before the chemical treatment leads to devastating results by hindering the chemical efficacy.
3. Provide the novelty of this review.
Response: The abstract is rewritten and an extensive change has been conducted in introduction section to reflect the novelty of this review.
4. Author Can include the future scope.
Response: Thanks for your suggestion. A future scope section has been added.
5. Author can include the summary either in the way of figure or table to explain the inference.
Response: Thanks for your comment. An extensive change has been made in the conclusion section where SDC and SPDC are analyzed thoroughly with new addition.
6. Discuss in detail about the elastic modulus and elongation of ABS-sawdust in detail, can compare it to other variants discussed.
Response: Thanks for your comment. Additional information is added there.
7. Provide reason for the flame-retardant loading condition as shown in figure 7.
Response: The flame-retardant loading condition investigated by Suoware et al. [45] demonstrates the trade-offs between improving tensile strength and maintaining tensile modulus in NaOH-treated sawdust composites. The addition of 12% ATH significantly enhances the tensile strength by 84% compared to composites without flame retardants. This improvement is attributed to the superior interfacial adhesion between ATH and the polyester resin, which facilitates effective stress transfer within the composite matrix. However, other flame retardant loadings, such as 12% APP-GAP and the hybrid 18% ATH/APP-GAP, result in lower tensile strengths (12.4 MPa and 16.5 MPa, respectively). These differences arise due to variations in the interaction of each flame retardant with the polyester matrix and sawdust, with ATH exhibiting better compatibility.
On the downside, all flame retardants negatively affect the tensile modulus, with composites containing 12% ATH, 12% APP-GAP, and 18% ATH/APP-GAP exhibiting lower values (1.50 MPa, 1.20 MPa, and 1.32 MPa, respectively) compared to the untreated composite’s 1.55 MPa. This reduction likely stems from the flame retardants acting as stress concentrators within the matrix, impeding the modulus despite enhancing strength in specific cases. These findings highlight that while ATH improves tensile strength due to its compatibility, the overall balance between strength and stiffness must be carefully considered in flame-retardant sawdust composites, as reflected in Figure 7.
8. Effect of treatment as shown in the SEM images 8 and 10 can be elaborated for clarity.
Response: Thanks for your suggestion. Necessary information added according to your instructions.
9. FTIR shown is explained technically.
Response: Thanks for your comment. A new paragraph with a technical discussion was added.

Reviewer 2 Report
Comments and Suggestions for Authors
Overall the paper is good. However, it needs to be improved:
1. Clarify the key contributions of the study in the abstract, focusing on the novelty and results.
2. Emphasize the significance of sawdust as a sustainable solution upfront and provide specific examples from previous research to establish a stronger context for this study.
3. The introduction lacks a strong problem statement and the clear motivation for using sawdust in polymer composites and concrete. Begin with a more impactful explanation of the environmental, economic, and engineering challenges related to traditional construction materials and sawdust disposal. This can help in building a compelling case for why the research was needed.
4. Clearly outline the objectives and scope of paper in the introduction to give readers a roadmap.
5.The current narrative mainly lists treatment methods and their impacts, but it lacks a comparative analysis across different studies and treatments. Add more critical analysis to discuss why certain treatments performed better or worse, the underlying mechanisms, and how they could be optimized.
6. Include a dedicated section summarizing key findings in a comparison table. This should focus on the effects of different treatments on mechanical, thermal, and durability properties.
7. Expand the discussion of the implications of using treated sawdust composites in real-world applications. There is minimal discussion on how these findings translate to practical uses in the construction industry or their potential for scaling up.
8. Figures 4 and 5 are referenced without discussing their implications thoroughly. Each figure should be interpreted in the context of the material's properties and compared with control samples to show the benefits or trade-offs of each treatment.
9. The conclusions are too general and repetitive. Rewrite the conclusion to emphasize the key contributions of this review and its potential impact on the field. Make concrete recommendations based on the findings.
Comments on the Quality of English LanguageImprove the english language by using english editors
Author Response
1. Clarify the key contributions of the study in the abstract, focusing on the novelty and results.
Response: Thanks for your comment. Additional information is added there.
2. Emphasize the significance of sawdust as a sustainable solution upfront and provide specific examples from previous research to establish a stronger context for this study.
Response: Thanks for your comment. Previous research observations and specific examples are added in introduction section.
3. The introduction lacks a strong problem statement and a clear motivation for using sawdust in polymer composites and concrete. Begin with a more impactful explanation of the environmental, economic, and engineering challenges related to traditional construction materials and sawdust disposal.
Response: An extensive change has been carried out in introduction section focusing environmental, economic, and engineering challenges related to traditional construction materials and sawdust disposal.
4. Clearly outline the objectives and scope of the paper in the introduction to give readers a roadmap.
Response: An extensive change to address the reviewer concern.
5. The current narrative mainly lists treatment methods and their impacts, but it lacks a comparative analysis across different studies and treatments. Add more critical analysis to discuss why certain treatments performed better or worse, the underlying mechanisms, and how they could be optimized.
Response: A comparative analysis against the figure 4, 5, 6,7,8,10 have been carried out. Additionally, comparative analysis between popular treatment and drawbacks are illustrated in conclusion sections. Additionally, future scope of this review is added after the conclusion.
6. Include a dedicated section summarizing key findings in a comparison table. This should focus on the effects of different treatments on mechanical, thermal, and durability properties.
Response: Table A1 and Table A2 are organized following mechanical and physical properties observation from different chemical treatments.
7. Expand the discussion of the implications of using treated sawdust composites in real-world applications. There is minimal discussion on how these findings translate to practical uses in the construction industry or their potential for scaling up.
Response: An extensive change has been carried out in introduction section.
8. Figures 4 and 5 are referenced without discussing their implications thoroughly. Each figure should be interpreted in the context of the material's properties and compared with control samples to show the benefits or trade-offs of each treatment.
Response: Thanks for your feedback. Necessary modification with additional information has been added.
9. The conclusions are too general and repetitive. Rewrite the conclusion to emphasize the key contributions of this review and its potential impact on the field. Make concrete recommendations based on the findings.
Response: A comparative analysis between popular treatment and drawbacks are illustrated in conclusion sections. Additionally, future scope of this review is added after the conclusion.
